# Modelling of Indicator *Escherichia coli* Contamination in Sentinel Oysters and Estuarine Water

**DOI:** 10.3390/ijerph16111971

**Published:** 2019-06-04

**Authors:** Saharuetai Jeamsripong, Edward R. Atwill

**Affiliations:** 1Research Unit in Microbial Food Safety and Antimicrobial Resistance, Department of Veterinary Public Health, Faculty of Veterinary Science, Chulalongkorn University, Pathumwan, Bangkok 10330, Thailand; 2Western Institute for Food Safety and Security, School of Veterinary Medicine, University of California, Davis, CA 95618, USA; ratwill@ucdavis.edu

**Keywords:** *Crassostrea*, *Escherichia coli*, estuarine water, fecal contamination, heavy metal contamination, log-transformation, *Salmonella*, *Shigella*

## Abstract

This study was performed to improve the ability to predict the concentrations of *Escherichia coli* in oyster meat and estuarine waters by using environmental parameters, and microbiological and heavy metal contamination from shellfish growing area in southern Thailand. Oyster meat (*n* = 144) and estuarine waters (*n* = 96) were tested for microbiological and heavy metal contamination from March 2016 to February 2017. Prevalence and mean concentrations of *E. coli* were 93.1% and 4.6 × 10^3^ most probable number (MPN)/g in oyster meat, and 78.1% and 2.2 × 10^2^ MPN/100 mL in estuarine water. Average 7-day precipitation, ambient air temperature, and the presence of *Salmonella* were associated with the concentrations of *E. coli* in oyster meat (*p* < 0.05). Raw data (MPN/g of oyster meat and MPN/100 mL of estuarine water) and log-transformed data (logMPN/g of oyster meat and logMPN/100 mL of estuarine water) of *E. coli* concentrations were examined within two contrasting regression models. However, the more valid predictions were conducted using non-log transformed values. These findings indicate that non-log transformed data can be used for building more accurate statistical models in microbiological food safety, and that significant environmental parameters can be used as a part of a rapid warning system to predict levels of *E. coli* before harvesting oysters.

## 1. Introduction

The consumption of raw seafood products, especially bivalves, poses potential risks for human health, since bivalves are effective filler feeders that can concentrate both nutrients and hazardous substances from the environment [1]. Bivalves can accumulate high bacterial loads and chemical contamination. Numerous seafood-borne outbreaks of *Salmonella, Shigella, Vibrio parahaemolyticus*, *Vibrio vulnificus, Streptococcus aureus* and *Clostridium botulinum* derived from oysters have been continuously reported globally [2,3,4,5]. The most common route of oyster-borne outbreaks has been traced to the consumption of raw or minimally-cooked oyster meat [6]. Heavy metal contamination in oyster meat has also been investigated in different locations. For example, in northern Vietnam, high levels of Zn, Cu, As, Cd, Pb and Cr were reported in rocky oysters, and in the north-central coast of Sinaloa in Mexico, the contamination of Zn, Cu, Cd, Pb, and Hg was observed in oyster meat intended for human consumption [7,8].

Laboratory microbiological analyses of pathogenic bacteria are generally expensive, difficult to perform and time-consuming; as a consequence, the determination of indicator bacteria i.e., fecal coliform and *Escherichia coli*, can be used as proxy to identify fecal contamination in seafood products and in harvested area [9]. In seafood safety monitoring programs, bivalves have been frequently used as a good environmental and biological indicator to identify contamination [10].

In field-based studies, average ambient air temperature, relative humidity, wind speed, seasons, geographical location, climate change, and aquatic animal captive approaches have been found to have had a strong influence on biological and chemical contamination of shellfish products [11]. Climate change is currently a global concern and it may contribute to a myriad of environmental factors such as heavy precipitation and temperature increase, influencing the persistence of bacterial impact on the dispersal of foodborne pathogens to the environment [12,13]. In oyster cultivation, main sources of fecal contamination from the environment to bivalve cultivation areas have been reported to come from domestic animals, wildlife, water runoff, birds, recreational sports, and sewage [14]. Nevertheless, it is challenging to identify the exact source of contamination to seafood products, which is often impractical, infeasible, and very difficult to perform. Therefore, monitoring and surveillance of environmental parameters can be adopted to predict bacterial accumulation in oyster meat and estuarine water.

Authorities from different countries such as the United States, the European Union (EU) and New Zealand have established national standards or guidelines regarding microbial contamination of shellfish, and criteria for determining the safety of shellfish intended for human consumption [15,16,17]. However, in many countries, including Thailand, no stringent guideline or standard has been strictly implemented with regard to the allowed level of *E. coli* contamination in fresh oysters for human consumption. Monitoring of environmental parameters, together with testing the concentrations of indicator bacteria of bivalves and estuarine waters in aquaculture growing areas, should be performed public health reasons. This information can also help farmers to identify the proper time to harvest oysters in order to decrease possible pathogenic bacterial contamination.

This study was conducted to improve the ability to predict the concentrations of *E. coli* in oyster meat and estuarine waters by using environmental parameters, the concentrations of total coliforms, fecal coliforms, and *V. parahaemolyticus*, the presence of *Salmonella* and *Shigella*, and levels of heavy metal contamination, including lead, manganese, and cadmium, in the Phang Nga area of southern Thailand. Many previous microbiological quality studies have primarily used log-transformed data to illustrate their results. Therefore, no log and log-transformed data of *E. coli* concentrations were used to generate and compare mixed-effect regression models of cultivated oysters and estuarine waters under tropical environmental conditions. The main objective of this study was to compare mixed regression models of oysters and estuarine waters, using non-log and log-transformed data to generate proper microbiological data.

## 2. Materials and Methods

### 2.1. Sample Collection

In this study, the data collection was conducted in aggregation with another experiment connected to a previous study [18]. Wild-caught natural oyster larvae, or spats, have been mainly used for oyster aquaculture in southern Thailand. These oyster larvae settle and attach to either man-made bamboo poles or motorcycle tires. When the oysters reach the age of 2–3 months, they are relocated to a grow-out structure cultivated area. The oyster shells are attached to one another on a nylon rope with cement. Oysters are raised under this hanging method until they reach the age of 10–12 months, which is provides market sized fresh oyster products.

A total of 240 samples derived from mature cultivated oysters (*Crassostrea lugubris* and *Crassostrea belcheri*) (*n* = 144) and estuarine waters (*n* = 96) were collected from March 2016 to February 2017 along the Phang Nga bay, Thap pud district, Krabi province in southern Thailand. Pooled oyster samples (*n* = 12) containing 10–12 market size oysters, and 500-mL samples of estuarine water (*n* = 8) were collected from four oyster cultivation sites each month for one year. All samples were kept in a cool box with ice pack during transportation. Following collection, oyster meat and estuarine water samples were processed immediately after arrival at the laboratory of the Department of Veterinary Public Health in Faculty of Veterinary Science in Chulalongkorn University.

### 2.2. Environmental Data Collection

Environmental parameters were measured, including average daily and weekly ambient air temperature (°C), relative humidity (%), maximum wind gust (m/s), wind speed (m/s), and precipitation (mm). Average daily environmental data was received on the day that samples were collected, and the data was recorded using a mobile anemometer wind meter (Kestrel 3000, Boothwyn, PA, USA). For average 7-day environmental parameters, all variable values were measured every three h/days, and then summarized. These 7-day environmental parameters were retrieved from the Thai meteorological department (https://www.tmd.go.th). Other factors, such as the presence of precipitation (present or absent), season (rain, summer, or winter), sampling month (month 1 to 12), and tidal condition (ebbing or flooding), were also recorded in this study.

### 2.3. Enumeration of Total Coliforms, Fecal Coliforms, E. coli and V. parahaemolyticus

The levels of total coliforms, fecal coliforms, *E. coli* and *V. parahaemolyticus* were quantified as most probable number (MPN) using a three-tube method, while the presence or absence of *Salmonella* and *Shigella* were tested in the oyster and estuarine water samples. Enumeration of total coliforms, fecal coliforms, and *E. coli* followed the United States Food and Drug Administration (U.S. FDA) Biological Analytical Manual (BAM) [19]. Briefly, 200 g of pooled oyster meat was blended at high speed for at least 30 s. Fifty grams of blended oyster meat were weighed and added to 450 mL of sterile phosphate buffered saline (PBS) (Difco, Sparks, MD, USA). For estuarine water, the sample was mixed with PBS at a 1:10 dilution. The remaining blended oyster meat and estuarine water were kept for further bacterial and heavy metal analyses. The mixture solution of each oyster and estuarine water sample was individually diluted in lactose broth (Difco) and incubated at 37 °C overnight. One loopful of individual lactose broth tube was transferred to brilliant green lactose bile (BGLB) (Difco) and incubated at 35 °C for 24 h. Then, positive tubes were recorded. To confirm fecal coliforms, a loopful of lactose broth tube was transferred to EC broth (Difco). After overnight incubation at 44.5 °C, the gas production was recorded as positive. For confirmation of *E. coli*, a loopful of EC broth was streaked onto Levine-eosin methylene blue (L-EMB) (Difco) agar plates. The suspected *E. coli* colonies were streaked on plate count agar (PCA) (Difco), and then confirmed using biochemical testing.

Measurement of concentrations of *V. parahaemolyticus* in pooled oyster meat and estuarine water samples also followed the U.S. FDA’s BAM method [20]. Briefly, mixture solution of the prepared samples from the previous step was serially diluted in alkaline peptone water (APW) (Difco) in three consecutive tubes to determine the populations of *V. parahaemolyticus* in oyster and estuarine water samples. The APW tubes were incubated at 37 °C overnight, and then a loopful from APW positive (turbid) tubes was streaked to thiosulfate-citrate-bile salts-sucrose (TCBS) (Difco) agar plate and confirmed by CHROMagar^TM^ Vibrio (HiMedia Laboratories Ltd., Mumbai, India) agar plate. Positive colonies were biochemically confirmed.

In determining bacterial loads in all samples from oyster and estuarine water, the concentrations of total coliforms, fecal coliforms. *E. coli*, and *V. parahaemolyticus* were reported in MPN/g or logMPN/g of oyster meat and MPN/100 mL or logMPN/100 mL of estuarine water.

### 2.4. Determination of Salmonella spp. and Shigella spp.

The detection of *Salmonella* and *Shigella* followed protocols as described by the U.S. FDA’s BAM method, with slight modification [21,22]. Twenty-five g of oyster meat was enriched with 225 mL of lactose broth (difco), and estuarine water was mixed with double strength lactose broth. The mixture solution of the samples was set at room temperature (25 °C) for 60 min. Then, the suspension was incubated at 35 °C overnight. Each mixture solution (0.1 mL) was transferred into 10 mL of Rappaport-vassiliadis (RV) (Difco). A loopful of the suspension was streaked onto xylose lysine deoxycholate (XLD) (Difco), MacConkey (Difco) and Hektoen enteric (HE) (Difco) agar plates. Biochemical testing was then used to confirm presumptive colonies of *Salmonella* spp. For all positive *Salmonella* isolates, serotyping on a slide agglutination assay followed the Kaufman-White scheme method, based on commercial antiserum (S&A Reagents Lab, Bangkok, Thailand) [23].

For *Shigella* spp. detection, briefly, a total of 25 g of blended oyster meat and 25 mL of estuarine water were separately enriched with *Shigella* and *Salmonella* broth (Difco). The mixture suspension was incubated at 35 °C overnight. A loopful of suspension was streaked onto MacConkey (Difco) agar plate. All positive colonies were confirmed using biochemical testing.

### 2.5. Determination of Heavy Metals

Concentrations of lead (Pb), cadmium (Cd) and manganese (Mn) in pooled oyster meat and estuarine water samples were analyzed following the guidelines of the Association of Analytical Communities [24]. Briefly, 5 g samples of blended pooled oyster meat were weighed out, and then dried over an oven. The sample was added to HNO_3_ concentrate (Merck, Washington, DC, USA). The mixture solution was dried over a hot plate to receive one ml of residue. For each estuarine water sample, a total of 200 mL of water was thoroughly mixed and filtered, passing through 11 µm filter paper (Whatman, Maidstone, UK). The filtered solution was added to 30 mL of HNO_3_ and set overnight at room temperature. The mixture solution was then dried, and adjusted to the final solution by adding distilled water to receive 25 mL. The final solution of all samples was then filtered, passing through 0.45 µm filter paper (Whatman) and kept in the refrigerator for further analysis. The concentrations of Pb, Cd, and Mn were quantified using the Atomic Absorption Spectrophotometry (AAS: Varian model AA280FS, Agilent, USA) in the Science and Technology Research Equipment Centre (STRE) at Chulalongkorn University.

### 2.6. Statistical Analyses

The concentrations of *E. coli* in both pooled oyster meat and estuarine water samples were used in two different regression models. The dependent variable of the levels of *E. coli* was separately calculated on raw, or non-transformed, data (MPN/g of oyster meat and MPN/100 mL of estuarine water) and log-transformed data (logMPN/g of oyster meat and logMPN/100 mL of estuarine water). Mixed-effect regression models were used to describe the association between the concentrations of *E. coli* in oyster meat or estuarine water samples, and average daily and 7-day environmental parameters, bacterial contamination of total coliforms and fecal coliforms, *Salmonella*, *Shigella*, and *V. parahaemolyticus*, and the levels of Pb, Mn and Cd. The *p*-values and C.I.’s of regression analyses for potential correlated data within the same location were adjusted by using a robust variance estimator. Univariate analysis and a backward elimination analysis were performed to build the multivariable mixed-effects regressions of pooled oysters and estuarine waters. The criteria used to select biological, chemical, and environmental factors affecting the concentrations of *E. coli* were based on potential biological meaning and AICs of individual models in order to select appropriate mixed-effect regression models. To perform all statistical analyses, Stata version 14 software (StatacCorp LP, College Station, TX, USA) was used.

## 3. Results

Average daily and 7-day environmental parameters on maximum wind gust (m/s), current wind speed (m/s), precipitation (mm), ambient air temperature (°C) and relative humidity (%) were summarized (Table 1). No log or log-transformed data were calculated and then used to compare within the same sample type. No log-transformed or raw data of the concentrations of *E. coli* of MPM/g of oysters (Model A) and log-transformed data of logMPM/g of oysters (Model B), and MPN/100 mL of estuarine water (Model C) and logMPN/100 mL of estuarine water (Model D) were then compared.

Under the univariate analysis, most environmental variables were found to be of significance (*p* < 0.05) in predicting the concentrations of *E. coli* in oyster meat and estuarine water, except the daily maximum wind gust in the Models C and D, and the daily current wind speed in the Models B, C, and D (Table 1).

In this study, the concentrations of *E. coli* ranged from 4.6 to 2.2 × 10^4^ MPN/g of pooled oyster meat, and 8.0 to 4.6 × 10^3^ MPN/100 mL of estuarine waters. The prevalence of *E. coli* was observed in 93.06% of oyster meat and 78.13% of estuary water. In this study, the lower limit of detection was approximately 5 MPN of *E. coli*/g of oyster meat and 8 MPN of *E. coli*/100 mL of estuarine water. The detection of *Shigella* and *Salmonella* were 7.64% and 30.56% of pooled oyster meat. Prevalence of *Shigella* was as high as 27.08% in 100 mL of estuarine waters, while no positive *Salmonella* was observed in pooled oyster meat samples. The positive *Salmonella* isolates from oyster meat samples were serotyped, and the results of this *Salmonella* serotyping showed that dominant serotypes were Paratyphi B 22.7% (*n* = 10/44) followed by Seremban 11.4% (*n* = 5/44) and Kentucky 9.1% (*n* = 4/11), respectively.

The univariate analysis of the concentrations of *E. coli* for variables associated with the concentrations of total coliforms, fecal coliforms, and *V. parahaemolyticus*, the presence of *Salmonella* and *Shigella* in the sample, and the levels of Mn, Pb and Cd are presented in Table 2. The concentrations of total coliforms and fecal coliforms, and the presence of *Shigella* were significant parameters in Models A, B, C and D (*p* < 0.05). The presence of *Salmonella* in the pooled oyster samples was associated with the concentrations of *E. coli* (Model A and B), whereas no *Salmonella* was detected from estuarine waters (Model C and D). The concentrations of *V. parahaemolyticus* were an insignificant parameter in all statistical models. For quantitative analysis of heavy metals, the levels of Mn were a significant predictor of the concentrations of *E. coli* in pooled oyster meat in model A (MPN/g) and Model B (logMPN/g). On the other hand, the concentrations of Cd and Pb were associated with the concentration of *E. coli* of estuarine water in Model C (MPN/100 mL) and Model D (logMPN/100 mL).

In Table 1 and Table 2, the significant variables with *p*-value < 0.05 from the univariate analyses of Models A, B, C, and D were included to perform multivariate analyses to predict the concentrations of *E. coli* in oyster meat and estuarine water. The dispersion of data can be calculated based on a log likelihood function test. The mixed-effects regression models of the concentrations of *E. coli* were transformed into logMPN/g of pooled oyster meat in Model B, and logMPN/100 mL of estuarine water in Model D. In the equation of a regression line, logMPN (y) was equal to a + bx, where y is outcome or dependent variable, a is the y-intercept, b is the slope, and x is predictor or independent variable. The concentrations of *E. coli* (MPN) = e^(a+bx)^ were shown in regression equations of pooled oyster meat (Model A) and estuarine water (Model C) under the mixed-effects negative binomial regression models.

The mixed-effects regression models between no log- and log-transformed data of concentrations of *E. coli* were compared in pooled oyster samples (Models A and B) and estuarine water samples (Models C and D), which are presented in Table 3 and Table 4, respectively. Prior 7-day average precipitation, average ambient air temperature, and the presence of *Salmonella* in the pooled oyster meat were significant variables included in Model A (no log-transformed data) and Model B (log-transformed data) to predict the concentrations of *E. coli* in pooled oyster meat samples under mixed-effects regression models (Table 3).

Similar results were observed in the mixed-effects regression models of estuarine water. Seven-day average precipitation and ambient air temperature were included in the final models (*p* < 0.05) of Model C (no log-transformed data) and Model D (log-transformed data) to predict the concentrations of *E. coli* in 100 mL of estuarine water (Table 4). No positive *Salmonella* was detected in estuarine water samples, so this variable was dropped from Models C and D. Other variables were also excluded from the final models due to non-significant variables (*p* > 0.05).

In the final regression models, a log scale of the original dataset, which was log of mean (Model A: MPN/g of oyster meat and Model C: MPN/100 mL of estuarine water) was compared with mean of log (Model B: logMPN/g of oyster meat and Model D: logMPN/100 mL of estuarine water) as a function of average ambient air temperature and 7-day precipitation of pooled oyster meat (Figure 1) and estuarine water (Figure 2).

In summary, average concentrations of *E. coli* in oyster and estuarine water samples were compared in different ranges of temperature (25–40 °C) and volume of precipitation (0–100 mm) (Table 5). The analysis of average concentrations of *E. coli* from no-log transformation data shows higher concentrations than the log-transformed data. For example, in oyster samples, the different concentrations of *E. coli* in a log scale range from 0.50 to 0.88 in ambient air temperature, and from 0.28 to 1.19 in levels of precipitation. In addition, the log concentrations of *E. coli* in estuarine water samples is observed to range from 0.73 to 1.04 in ambient air temperature, and from 0.06 to 0.88 in levels of precipitation. Hence, the log of mean (Models A and B) predicts higher concentrations of *E. coli* than the mean of log (Models B and D). The log of mean provided proper estimation of the concentrations of *E. coli* due to the lower residue observed in the log of mean than the mean of log.

## 4. Discussion

The ability to predict concentrations of *E. coli* in pooled oyster meat and estuarine water samples by using average daily and 7-days of environmental parameters, microbiological contamination and the levels of heavy metals was investigated in this study. These variables were summarized, and then univariate analyses were performed, as summarized in Table 1 and Table 2. Seven-day average precipitation and ambient air temperature were significant parameters (*p* < 0.05) in the final regression models used to predict the concentrations of *E. coli* in oysters and estuarine waters. In mixed-effect regression models, the significant parameters from no log- and log-transformed data of the concentrations of *E. coli* are displayed in Models A, B, C, and D (Table 3 and Table 4). Other non-significant variables (*p* > 0.05) such as maximum wind gust, current wind speed, the concentrations of *V. parahaemolyticus*, and the levels of heavy metal were dropped from the final regression models. The lack of significance among these variables is possibly due to lack of variability of the data presented during sample collection. For example, one-day current wind values ranged between 0 and 1.5 (SD ± 0.4) m/s, which was typically constant over the 12-month period of the study. Additionally, average 7-day current wind speed had a mean of 1.0 m/s with SD 0.4 m/s, values which were also consistent over time. This finding was similar to the consistency of the concentrations of Mn, Pb and Cd observed in all samples from oyster meat and estuarine water. Hence, these consistent variables were dropped from the final regression model due to their homogeneity. Thus, fluctuating environmental parameters are the only ones found to be of significance in predicting concentrations of *E. coli* contamination in oysters and estuarine waters.

In our final regression models, 7-day averages for precipitation and daily ambient air temperature were significant predictors for estimating the concentrations of *E. coli* in pooled oyster meat observed in Models A and B, and estuarine water samples observed in Models C and D. The levels of precipitation were also positively associated with the concentrations of *E. coli*, while a negative relationship was observed between ambient air temperature and the concentrations of *E. coli* in pooled oyster meat and estuarine water samples. These findings are supported by the study from Scopel et al. which found that, on average, the accumulation of *E. coli* in seawater increased up to 2 log colony forming unit (CFU) with the observed highest concentration at 4500 CFU/100 mL after precipitation at a beach [25]. A positive correlation between the impact of precipitation and concentrations of *E. coli* was also observed among blue mussels [26]. This study suggests that precipitation can introduce microbial contaminants to coastal areas and other locations of oyster cultivation. One possible explanation would be run off water from nearby communities and discharges from municipal water may wash fecal contamination into oyster cultivating areas. During periods of higher precipitation, there is a greater likelihood of detecting *E. coli* in both oyster meat and estuarine water. This finding is supported by many studies, which is reinforced by the fact that waterborne disease outbreaks have been increasingly reported during periods of rainfall, where heavy rain-washed microbial contaminants into areas of oyster cultivation [27,28].

Considering the temperature parameters, a negative relationship was observed in this study between ambient air temperature and the concentrations of *E. coli*. Other published papers support the suggestion that the rate of inactivation of *E. coli* is positively associated with temperature [29]; in addition, tendency towards high bacterial accumulation in clams was observed when the temperature increased [30]. The optimal temperature of maximal clearance of *Ostrea edulis* oysters falls in a temperature range between 12 °C and 19 °C (Šolić et al., 1999). This study found that temperature and precipitation are key factors that influence the survival of *E. coli* in oyster meat and estuarine water.

Models of log-transformed data have been regularly used for predicting of bacterial inactivation and persistence in the environment, as well as food commodities [31,32]. In this study, the concentrations of *E. coli* were normalized by using log transformation, and then regressed by significant variables, as presented in Model B of pooled oyster meat (logMPN/g of oyster) and Model D of estuarine water (logMPM/100 mL of estuarine water). The value of log concentrations of *E. coli* in models B and D was lower than no-log transformation of *E. coli* in models A and C when the log-transformed data was calculated back to raw or original data (Figure 1 and Figure 2). Our study elaborated that log transformed data of bacterial concentration underestimates predicted values of *E. coli* contamination of oyster meat and estuarine water samples. This finding on the normalized data could introduce a bias for bacterial estimation according to Jensen’s inequality theorem [33]. Jensen’s equality is explained by an equation in which the log of mean is greater than or equal to mean of log. Moreover, our results showed that the no-log transformation data provides more accurate estimation on the concentrations of *E. coli* in both pooled oyster meat and estuarine water samples.

Subsequently, mixed-effects negative binomial regression models were performed to predict the concentrations of *E. coli* in Model A of oyster and Model C of estuarine water. The negative binomial regression model is suitable for overly dispersed count data, and this method has been widely used in microbiological studies, such as pre-harvest contamination of indicator and pathogenic bacteria in mixed produce, and the impact of farm management and environmental factors on the level of indicator bacteria in spinach [34,35,36]. Furthermore, more advanced studies on diversity of microbial ecological interaction networks on metagenomic data and microbiome count data have also used negative binomial regression to generate proper statistical models [37,38,39]. However, many microbiological studies use log transformation to exhibit the distribution of bacteria in various food commodities due to the fact that it is easy to perform when diversity of bacteria is observed.

The log-transformed data provided lower estimates of concentrations of *E. coli* after being transformed to raw data, compared to no log-transformation. For example, with average ambient air temperature of 28 °C and 7-day average levels of precipitation at 19.11 mm, the mean of log was 3.59 (Model B) and the observed log of mean was 4.05 (Model A). The log difference between a log of mean and a mean of log concentration of *E. coli* was 0.46 log (4.05 − 3.59 = 0.46), which is presented in Figure 1. More specifically, each additional degree Celsius of ambient air temperature is associated with a 31% (10^−0^^.16^ = 0.69) decrease in the concentration of *E. coli* when transformed to raw data in model B. A higher rate of reduction in the concentration of *E. coli* is observed in model B, as compared with model A. For example, each additional degree Celsius of air temperature in model A is related to a 24% (e^−0^^.28^ = 0.76) reduction in the concentration of *E. coli* (Table 3).

A similar result was seen with the 7-day average level of precipitation, with each additional millimeter of average precipitation in Model B associated with a 4% (10^−0^^.0156^ = 0.96) decrease in the concentration of *E. coli.* Each additional millimeter of 7-day average precipitation in Model A, by contrast, is associated with a 2% (e^−0^^.0229^ = 0.98) reduction in *E. coli* in pooled oyster meat (Table 3). Even though the difference between the concentrations of *E. coli* in Models A and B was not great, both models differ substantially in their predictions, as shown in Figure 1. Interestingly, despite the fact that these bacteria are considered to be pathogens, few bacterial cells lead to serious complications with regard to human health.

In Models C and D, the concentrations of fecal coliforms, average ambient air temperature and 7-day precipitation were significant parameters for predicting the concentrations of *E. coli* in estuarine water samples. In Model D, each additional degree (Celsius) of ambient air temperature is associated with a 15% (10^−0^^.07^ = 0.85) reduction in the concentration of *E. coli* when transformed to raw, or original, data. In Model C, a 10% (e^−0^^.11^ = 0.90) decrease in the concentration of *E. coli* is associated with this same increase in temperature (Table 4). Similarly, focusing on the level of precipitation, each additional of millimeter of 7-day average precipitation is associated with a 4% (10^−0^^.0199^ = 0.96) decrease in the concentrations of *E. coli* in Model D, whereas a 3% (e^−0^^.0281^ = 0.97) reduction in the concentrations of *E. coli* is observed in the Model C (Table 4). Fecal coliforms are a significant variable for estimating the concentration of *E. coli* in estuarine water, although the rate of reduction in the concentration of *E. coli* was similar in Model C (e^0^^.000875^ = 1.0) and Model D (10^−0^^.000243^ = 1.0). Even though these significant variables are not greatly different between models C and D, the intercept coefficients showed largely dissimilar values, as displayed in Table 4. Using the averages of all significant variables, the log concentrations of *E. coli* were at approximately 1.74 and 1.47 logMPN/100 mL of estuarine water in Models C and D, respectively. This result indicates that log-transformed data provides a lower prediction of *E. coli* populations compared with no-log transformed data.

Concentrations of indicator bacteria have varying ranges, depending on geographical location, environmental parameters, and the presence of pathogenic bacteria. For example, the concentrations of total and fecal coliforms in oyster meat harvested in Korea were detected from 65.6 to 127.1 MPN/100 g of total coliforms and 23.7 to 26.7 MPN/100 g of fecal coliforms, respectively [40]. The finding was supported by a study from Mok et al. that showed similar concentrations of fecal coliforms as in Korea. The concentrations of *E. coli* can be used to estimate the pathogenic bacterial contamination of shellfish. The low levels of *E. coli* load in shellfish indicates that *Salmonella* contamination may be absent from these shellfish [41]. The presence of indicator bacteria of bivalves suggests that the environment is polluted, and aquatic animals and their products are possibly contaminated.

The limitations of this study include the limited ranges of the observed parameters such as temperature (between 25 °C and 40 °C) and the level of precipitation (which did not exceed 100 mm). Thus, the prediction of the concentrations of *E. coli* should be estimated and interpreted within the provided ranges observed in this study. This finding cannot generalize about outside ranges to predict the concentration of *E. coli*. This study suggests that monitoring of environmental factors associated with the concentrations of *E. coli* in oysters and estuarine waters should be performed regularly to decrease the risk of undetected seafood contamination. Furthermore, use of no-log transformation will provide better estimates and less bias than log transformed data.

## 5. Conclusions

Fresh oysters and estuarine water can become contaminated with non-pathogenic and pathogenic bacteria of public health significance. Environmental factors used to predict the concentrations of such bacteria should provide a useful tool for detecting non-pathogenic bacteria, and possibly identifying pathogenic bacterial contamination before oysters are harvested. Certain environmental parameters such as precipitation and ambient air temperature were found to be significantly associated with the concentrations of *E. coli* in oyster meat and estuarine water samples. To conclude, no-log transformed data should be utilized, rather than log-transformed data, to predict the concentrations of *E. coli* in order to achieve the most precise estimation. This piece of information can also be applied to other quantitative microbiological studies to enhance precision and reduce the bias of the studies.

## Figures and Tables

**Figure 1 ijerph-16-01971-f001:**
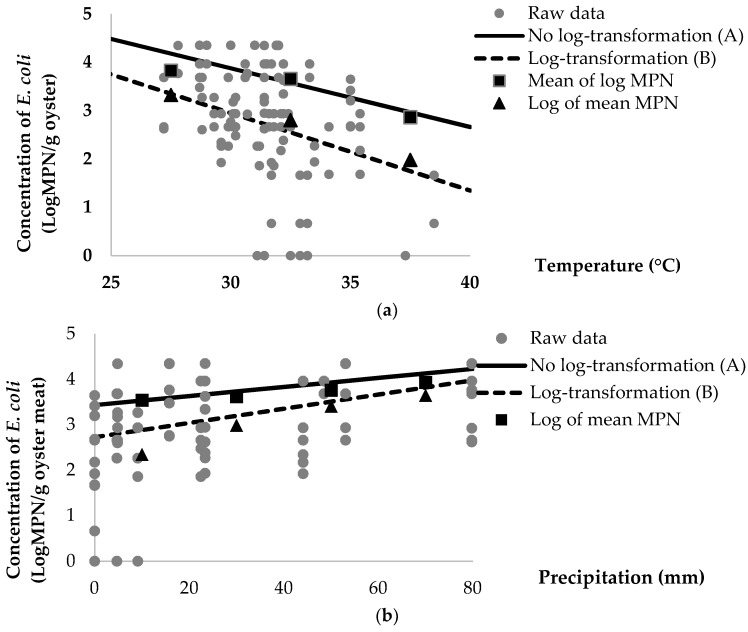
Predicted average population of *E. coli*/g of pooled oyster meat as a function of (**a**) ambient air temperature and (**b**) 7-day average precipitation between no log-transformed (Model A) and log-transformed (Model B) data.

**Figure 2 ijerph-16-01971-f002:**
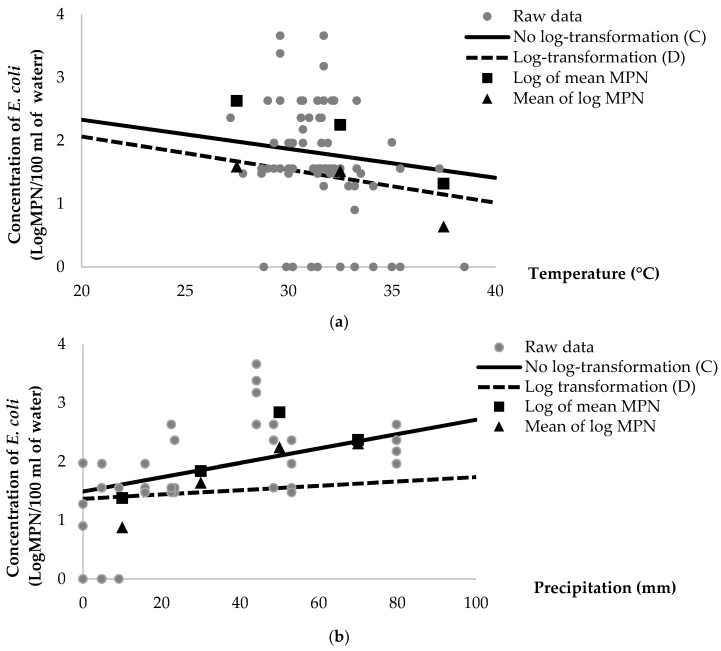
Predicted average population of *E. coli*/g of estuarine water as a function of (**a**) ambient air temperature and (**b**) 7-day average precipitation between no log-transformed (Model C) and log-transformed (Model D).

**Table 1 ijerph-16-01971-t001:** Summary of 1-day and 7-day data for descriptive statistics on environmental parameters collected by Thai metrological department station prior to collection of oysters and estuarine waters over 12 months from March 2016 to February 2017.

Parameter	Med	SD	Min	Max	Pooled Oyster (*p*-Value ^1^)			Estuarine Water (*p*-Value ^1^)
					Model A ^2^	Model B ^3^	Model C ^2^	Model D ^3^
**1-day environmental data**								
Maximum wind gust (m/s)	6.2	2.4	5.2	12.9	0.001	<0.0001	0.968	0.313
Current wind speed (m/s)	0.8	0.4	0.0	1.5	0.002	0.066	0.713	0.383
Precipitation (mm)	2.4	28.8	0.0	87.8	<0.0001	0.001	<0.0001	<0.0001
Temperature (°C)	28.0	1.5	25.2	29.7	0.020	<0.0001	0.008	<0.0001
Relative humidity (%)	83.0	7.8	74.0	97.0	<0.0001	<0.0001	<0.0001	<0.0001
**7-day environmental data**								
Maximum wind gust (m/s)	6.6	1.9	5.6	11.4	<0.0001	<0.0001	<0.0001	<0.0001
Current wind speed (m/s)	1.0	0.4	0.6	1.8	<0.0001	<0.0001	<0.0001	<0.0001
Precipitation (mm)	19.1	24.4	0.0	79.8	<0.0001	<0.0001	<0.0001	<0.0001
Temperature (°C)	27.3	1.0	26.1	29.6	<0.0001	<0.0001	<0.0001	<0.0001
Relative humidity (%)	86.1	4.9	78.3	92.7	<0.0001	<0.0001	<0.0001	<0.0001

^1^*p*-values were adjusted for potential intra-group correlation within sampling locations; ^2^ No log of *E. coli* (MPN/g of oyster meat or MPN/100 mL of estuarine water); ^3^ LogMPN of *E. coli* (LogMPN/g of oyster or LogMPN/100 mL of estuarine water; Med: Median; SD: Standard Deviation; Min: Minimum; Max: Maximum.

**Table 2 ijerph-16-01971-t002:** Univariate analysis of the concentrations of *E. coli* for factors associated with bacterial and chemical contamination of pooled oysters and estuarine waters.

Parameter	Pooled Oyster (*p-*Value ^1^)		Estuarine Water (*p-*Value ^1^)	
	Model A ^2^	Model B ^3^	Model C ^2^	Model D ^3^
Concentrations of bacteria ^4^				
TC concentration	<0.0001	<0.0001	<0.0001	<0.0001
FC concentration	<0.0001	<0.0001	<0.0001	<0.0001
VP concentration	0.224	0.116	0.012	0.650
The present of *Salmonella*				
No (reference)	-	-	-	-
Yes	<0.0001	<0.0001	*	*
The present of *Shigella*				
No (reference)	-	-	-	-
Yes	<0.0001	0.006	<0.0001	<0.0001
Heavy metal (ppm)				
Mn	0.020	0.002	0.729	0.343
Cd	0.393	0.718	<0.0001	<0.0001
Pb	<0.0001	0.382	<0.0001	<0.0001

^1^*p*-values were adjusted for potential intra-group correlation within sampling locations; ^2^ No-log data of *E. coli*; ^3^ LogMPN of *E. coli*; ^4^ Units of bacterial concentration is MPN/g of oyster (Model A), logMPN/g of oyster (Model B), MPN/100 mL of water (Model C) and logMPN/100 mL of water (Model D); * No positive *Salmonella* observed in estuarine water samples; TC: Total coliforms; and FC: Fecal coliforms.

**Table 3 ijerph-16-01971-t003:** Mixed-effects regression models between no log- and log-transformed data of the concentrations of *E. coli* collected from pooled oyster from March 2016 and February 2017.

Factor	Coefficient	95% CI ^1^	*p-*Value ^1^
**Model A: No log-transformation**			
Intercept	16.11	7.38 to 24.83	<0.0001
Precipitation prior 7 days (mm)	2.29 × 10^−2^	1.36 × 10^−3^ to 3.22 × 10^−4^	<0.0001
Temperature (°C)	−0.28	−0.54 to −0.02	0.036
Salmonella present in sample			
No	0.0	-	-
Yes	0.62	0.01 to 1.23	0.046
**Model B: Log-transformation**			
Intercept	7.36	5.04 to 9.67	<0.0001
Precipitation prior 7 days (mm)	1.56 × 10^−2^	8.72 × 10^−3^ to 2.26 × 10^−2^	<0.0001
Temperature (°C)	−0.16	−0.23 to −0.09	<0.0001
Salmonella present in sample			
No	0.0	-	-
Yes	0.42	0.07 to 0.78	0.019

^1^ The 95% confidence interval (CI) and *p*-values were adjusted for potential intra-group correlation within oyster sampling locations.

**Table 4 ijerph-16-01971-t004:** Mixed-effects regression models between log- and log-transformed data of the concentration of *E. coli* collected from estuarine water samples from March 2016 to February 2017.

Factor	Coefficient	95% CI ^1^	*p*-Value ^1^
**Model C: No-log transformation**			
Intercept	6.40	3.30 to 9.49	<0.0001
Concentration of FC (MPN/100 mL)	8.75 × 10^−4^	5.88 × 10^−^^4^ to 1.16 × 10^−^^3^	<0.0001
Precipitation prior 7 days (mm)	2.81 × 10^−2^	2.28 × 10^−^^2^ to 3.33 × 10^−^^2^	0.030
Temperature (°C)	−0.11	−0.20 to −0.01	<0.0001
**Model D: Log transformation**			
Intercept	2.95	0.89 to 5.01	0.005
Concentration of FC (logMPN/100 mL)	2.43 × 10^−4^	1.47 × 10^−^^4^ to 3.40 × 10^−^^4^	<0.0001
Precipitation prior 7 days (mm)	1.99 × 10^−^^2^	1.45 × 10^−^^2^ to 2.53 × 10^−^^2^	<0.0001
Temperature (°C)	−0.07	−0.13 to −2.63 × 10^−^^2^	0.041

^1^ The 95% confidence interval (CI) and *p*-values were adjusted for potential intra-group correlation within sampling locations; and FC, Fecal coliform.

**Table 5 ijerph-16-01971-t005:** Summary of average concentrations of *E. coli*/g of oyster meat and *E. coli*/100 mL of estuarine water samples in different ranges of temperature (°C) and 7-day precipitation averages (mm) from March 2016 to February 2017.

Sample	Parameter	Number of Sample (%)	Mean of Log (MPN/g or MPN/100 mL)	Log of Mean (MPN/g or MPN/100 mL)	Log Difference ^1^
Pooled oyster	Temperature				
	25.0–29.9	36 (25.0%)	3.82	3.32	0.50
	30.0–34.9	90 (62.5%)	3.65	2.80	0.85
	35.0–39.9	18 (12.5%)	2.86	1.98	0.88
	Total	144 (100%)	3.66	2.83	0.83
Estuarine water	Temperature				
	25.0–29.9	20 (20.8%)	2.63	1.59	1.04
	30.0–34.9	68 (70.8%)	2.25	1.52	0.73
	35.0–39.9	8 (8.30%)	1.32	0.64	0.68
	Total	96 (100%)	2.34	1.46	0.88
Pooled oyster	Precipitation				
	0–19.9	72 (50.0%)	3.54	2.35	1.19
	20.0–39.9	24 (16.7%)	3.62	2.99	0.63
	40.0–59.9	36 (25.0%)	3.76	3.41	0.35
	60.0–79.9	12 (8.3%)	3.93	3.65	0.28
	Total	144 (100%)	3.66	2.83	0.83
Estuarine water	Precipitation				
	0–19.9	48 (50.0%)	1.38	0.88	0.50
	20.0–39.9	16 (16.7%)	1.84	1.64	0.20
	40.0–59.9	24 (25.0%)	2.84	2.24	0.60
	60.0–79.9	8 (8.3%)	2.37	2.31	0.06
	Total	96 (100%)	2.34	1.46	0.88

^1^ The log difference between mean of log (logMPN/g of oyster meat and logMPN/100 of estuarine water) and log of mean concentration (MPN/g of oyster meat and MPN/100 mL of estuarine water) of *E. coli.*

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
