# Peer review of "Modelling of Indicator Escherichia coli Contamination in Sentinel Oysters and Estuarine Water"

_ijerph, 2019, doi:10.3390/ijerph16111971_

Round 1

Reviewer 1 Report

Journal: Int. J. Environ. Res. Public Health

Manusctipt Title: Modelling of Indicator E. coli Contamination in Sentinel Oysters and Estuarine Water

Manuscript number: ijerph-502281

In this manuscript, a study on improvement of ability to predict the concentration of bacteria of the species Escherichia coli in oyster meat and estuarine waters by using environmental parameters, and microbiological and heavy metal contamination from shellfish growing area in southern Thailand, was carried out.

This study was conducted in order to evaluate contaminaion with non-pathogenic and pathogenic bacteria of public health significance, in fresh oysters and estuarine water.

The environmental factors used to predict the concentration of such bacteria should provide a useful tool for detecting non-pathigenic bacteria, and possibly identifying pathogenic bacterial contamination before harvesting oysters.

According to results obtined in this study, certain environmental parameterssuch as precipitation and ambient air temperature were found to be significantly associated with the concentrations of E. coli in oyster meat and estuarine water samples.

Evidences from elaboration of the obtained data, highlighted that no-log transformed data should be utilized, instead of log-transformed data, in order to obtain the most precise estimation on the concentration of E. coli.

These estimations can also be applied to other quantitative microbiological studies to enhance precision and reduce eventual bias of the studies.

The manuscript is innovative in introducing a different point of view in treating detection of contamination by pathogenic and non-pathogenic bacteria, by using bacteria of the species Escherichia coli as bioindicators. The novelty resides in the use of environmental parameters, and microbiological and heavy metal contamination from a shellfish growing area to predict microbiological contamination.

Considerations on non-log and log-transformed data to generate proper microbiological data, strongly suggests that non-log transformed data can be used for building more accurate statistical models in microbiological food safety and that significant environmental parameters can be used in order to obtain rapid information on prediction levels of E. coli before harvesting oysters.

This type of approach is extremely interesting as it can give insights from the microbiological points of view, closely related to other aspects such as environmental parameters and heavy metal contamination.

This approach will also provide important information to investigate any microbial adaptations to these aspects, to be used as a bioindicator, if necessary. For example, the development of investigation of genes for bacterial resistance to heavy metals can represent a synergistic system to trace the distribution of bio-available metals, capable of interacting with bacteria, as the first levels of the trophic chain.

Revisions

Page 1 line 2 and line 12: “E. coli” change to “Escherichia coli”;

Page 3 line 39: “spp.” not in Italic style;

Page 9 lines 5 and 6: “Table 3” change to “Table 5”.

Author Response

Response to Reviewer and Revision to IJERPH

Dear Editor, 

The authors thank you for consideration of our manuscript (ID: ijerph-502281) entitle of Modelling of Indicator Escherichia coli Contamination in Sentinel Oysters and Estuarine Water. Our point-by-point responses to the reviewerscomments are described below.

Sincerely yours,

Saharuetai Jeamsripong

Response to Reviewer 1

1.     Page 1 line 2 and line 12: “E. coli” change to “Escherichia coli”;

The word “E. coli” was changed to “Escherichia coli”.

2.     Page 3 line 39: “spp.” not in Italic style;

The error has been corrected.

3.     Page 9 lines 5 and 6: “Table 3” change to “Table 5”.

The error has been corrected.

Reviewer 2 Report

The manuscript describes used of statistical methods to correlate various environmental and bacteriological measurements with E coli which is a standard bacterial for measuring the sanitary quality of shellfish and their growing waters.

The manuscript does not appear to make any startling revelations regarding the potential presence of pathogenic fecal bacteria like salmonella and shigella.  Excessive rainfall is known to be a key criteria for causing elevated fecal coliform levels in shellfish and water.  In fact mandatory closures for three weeks are typical in certain jurisdictions.  The “positive” correlation with water temp is a little curious.  But if one reads the manuscript carefully it is evident that they means from 25 C and above.  Presumably bacteria metabolisms, dissolved oxygen and perhaps oyster pumping metabolism are all factors relating to this.

the methods used for modeling are not well described the manuscript should provide more information about the models, equations or computer programs used

Specific comments

Paragraph 3 of the introduction seems to blame global warming.  This association seems pretty tenuous.  If global warming causes an acute impact on Thailand’s weather patterns particularly rainfall, I could see this connection. But in the absence of some objective data, this reviewer is skeptical about that statement.

Page 2 line 18 “allowed level of bacterial contamination”  please elaborate as to which bactrria or bacterial species you are speaking about.

Page 6 line 1  change 104 to 104

Page 6 lines 2-5; Prevalence…can you give the detection limits for the bacteria tested?   Not finding E coli in 100% of the oyster samples is surprising.  This suggest the testes were not very sensitive.

Author Response

Response to Reviewer and Revision to IJERPH

Dear Editor, 

The authors thank you for consideration of our manuscript (ID: ijerph-502281) entitle of Modelling of Indicator Escherichia coli Contamination in Sentinel Oysters and Estuarine Water. Our point-by-point responses to the reviewerscomments are described below.

Sincerely yours,

Saharuetai Jeamsripong

Response to Reviewer 2

Reviewer comments

Response to Reviewer 2

1. the methods used for modeling are not well described   the manuscript should provide more information about the models, equations or   computer programs used

Models and equations were addressed in lines 1-3 on page 7,   and the statistical program used in this study was indicated in statistical   analysis on page 4.

2. Paragraph 3 of   the introduction seems to blame global warming.  This association seems pretty tenuous.  If global warming causes an acute impact on   Thailand’s weather patterns particularly rainfall, I could see this   connection. But in the absence of some objective data, this reviewer is   skeptical about that statement.

The sentence is modified in lines 6-8 on page 2 as “Climate   change is currently a global concern and it may suggest as part of a myriad   of environmental factors such as heavy precipitation and temperature increase   influencing the persistence of bacterial impact on the dispersal of foodborne   pathogens to the environment [12-13]”.

3. Page 2 line 18 “allowed level of bacterial   contamination” please elaborate as to which bacteria or bacterial species you   are speaking about.

The bacteria are indicated as indicator bacteria such as E.   coli. The sentence was modified as “However, in many countries, including   Thailand, no stringent guideline or standard has been strictly implemented   with regard to the allowed level of E. coli contamination in fresh   oysters for human consumption.”.

4. Page 6 line 1 change 104 to 104

The error has been corrected.

5. Page 6 lines 2-5; Prevalence…can   you give the detection limits for the bacteria tested? Not finding E coli in   100% of the oyster samples is surprising.    This suggest the testes were not very sensitive.

The detection limits are added in lines 3-4 on page 6. In   this study, the lower limit of detection was approximately 5 MPN of E.   coli/g of oyster meat and 8 MPN of E. coli/100 ml of estuarine   water.
